# A portable feedback-controlled pump for monitoring eye outflow facility in conscious rats

Youssef Mohamed[1], Christopher L. Passaglia[1,2]*

1 Department of Medical Engineering, University of South Florida, Tampa, FL, United States of America,
2 Department of Ophthalmology, University of South Florida, Tampa, FL, United States of America

* passaglia@usf.edu

**Data Availability Statement:** All relevant data are within the paper and its Supporting Information files.

**Funding:** CLP acknowledges the National Institutes of Health grant R01 EY027037 (nih.gov). The

## Abstract

Intraocular pressure (IOP) is heavily influenced by the resistance of trabecular outflow pathways through which most of the aqueous humor produced by the eye continuously drains. The standard method of quantifying outflow resistance and other aspects of ocular fluid dynamics is eye cannulation, which allows for direct measurement and manipulation of IOP and flow in animal models. Since the method is invasive, indirect techniques that are slower and less accurate must be used for chronological studies. A novel technology is introduced that can autonomously measure outflow facility in conscious rats multiple times a day. A smart portable micropump infuses fluid into the eye through a permanently-implanted cannula and dynamically adjusts flow rate using a unique proportional feedback algorithm that sets IOP to a target level, even though IOP fluctuates erratically in awake free-moving animals. Pressure-flow data collected by the system from anesthetized rats were validated against intraocular recordings with commercial pressure and flow sensors. System and sensor estimates of outflow facility were indistinguishable, averaging $23 \pm 3$ nl·min$^{-1}$·mmHg$^{-1}$ across animals (n = 11). Pressure-flow data were then collected round-the-clock for several days from conscious rats, while outflow facility was measured every few hours. A significant diurnal facility rhythm was observed in every animal (n = 4), with mean daytime level of $22 \pm 10$ nl·min$^{-1}$·mmHg$^{-1}$ and mean nighttime level of $15 \pm 7$ nl·min$^{-1}$·mmHg$^{-1}$. The rhythm correlated with diurnal changes in IOP and likely contributed prominently to those changes based on the day-night swing in facility magnitude. Hence, the portable smart pump offers a unique tool for repeated long-term monitoring of outflow facility and other possible parameters of ocular health. It could also be useful in animal glaucoma studies for reversibly inducing acute or chronic ocular hypertension without explicitly damaging trabecular outflow pathways.

## Introduction

Eye health depends on a delicate balance of ocular fluid influx and efflux. Excess fluid volume raises intraocular pressure (IOP), which can cause glaucomatous nerve damage and vision loss

funders had no role in study design, data collection and analysis, decision to publish, or preparation of the manuscript.

**Competing interests:** I have read the journal's policy and the authors of this manuscript have the following competing interests: U.S. patents 9022968, 9314375, and 10758408. This does not alter our adherence to PLOS ONE policies on sharing data and materials.

if large in amplitude or long in duration, while a shortfall in volume lowers IOP, which can cause retinal wrinkling and distorted vision [1–3]. Ocular fluid dynamics are driven primarily by changes in blood flow and aqueous humor flow. The former causes IOP to pulsate with each breath and heartbeat and shift with postural alterations in body center of gravity [4, 5]. The latter sets baseline IOP as the bulk of aqueous humor exits the eye through the trabecular meshwork, Schlemm's canal, and episcleral vessels that present outflow resistance. Aqueous production and outflow are also subject to autonomic modulation from stress, circadian rhythms, and other internal processes that cause IOP to fluctuate erratically from moment-to-moment and rise-and-fall daily [6–8].

Ocular fluid dynamics are tricky to quantify in live animals. Noninvasive methods like tonography and fluorophotometry can provide qualitative estimates of outflow facility, but they require cooperative subjects, biological assumptions, and long measurement times as fluid clearance is slow [9]. For more direct and accurate estimates, the eye must be cannulated with a tube or needle. Outflow facility can then be estimated via a constant flow (CF) or constant pressure (CP) perfusion technique. With the CF technique, fluid is infused into the eye at known flow rates and the steady-state IOP level is recorded for each rate [10, 11]. With the CP technique, IOP is set at different levels using a variable-height fluid reservoir [12] or feedback-controlled pump [13–15] and the steady-state flow rate needed to maintain each setpoint is recorded. Unfortunately, invasive methods like eye cannulation cause acute trauma and pose risk for chronic damage. They also require anesthesia, which can confound results by altering IOP and outflow facility [16, 17]. The duration and frequency of measurements is thereby limited and experiments are generally terminal.

In this paper we present the design, testing, and usage of an eye perfusion system for monitoring ocular fluid dynamics in awake freely-moving animals. The system is compact and lightweight, allowing for portability and cage-side placement. It also offers a low-cost option to the expensive bulky laboratory setups typically used on anesthetized animals. The system continuously records IOP via a cannula implanted permanently in the eye. Whenever the user desires, the system infuses fluid at a specified flow rate or sets IOP at a specified level by autoregulating flow rate. Outflow facility can thereby be measured at any time of day without need of anesthesia or repeated eye cannulation. System design is optimized for rat eyes and rat IOP variability, but design parameters could be adapted for use in larger animals.

## Materials and methods

### System design

Fig 1A shows a picture of the portable eye perfusion system connected by tubing to a 15-ml vial of saline. System electronics are housed in a custom plastic box about a printer ink cartridge in size (weight: 40 g) and interface with a computer via a USB cable. Fig 1B shows a picture of the experimental setup for monitoring outflow facility in conscious rats. The system is fluidically coupled to the eye via a tether that connects to a head mount affixed to the skull. Fig 1C illustrates system components. A microfluidic pump (mp6-hyb, Bartels mikrotechnic, Dortmund, Germany) draws fluid from a reservoir of balanced salt solution (BSS®, Alcon Laboratories, Geneva, Switzerland) and pushes it through a flow restrictor (PEEK tubing, outer and inner diameter: 360 and 50 μm, IDEX Health and Science, Oak Harbor, WA, USA) and connector tubing to a cannula inserted in the anterior chamber of the eye. A pressure sensor (TBPDANS005PGUCV, Honeywell, NC, USA) connected to tubing by a tee junction measures IOP and sends the signal to a microcontroller. The microcontroller digitizes the IOP signal and adjusts pump operation based on the measured pressure drop across the flow restrictor to either infuse fluid at a user-specified rate (CF mode) or set IOP at a user-specified

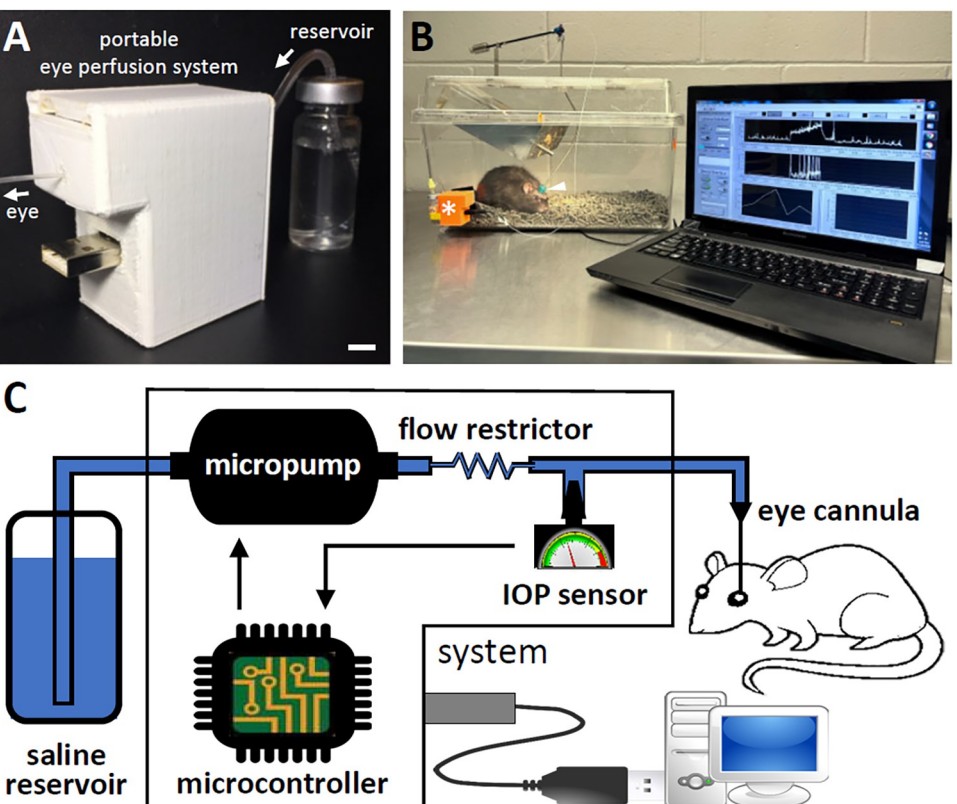

**Fig 1. Portable eye perfusion system.** (A) Image of the device connected to an external fluid reservoir. Scale bar: 5 mm. (B) Image of experimental setup in action on a conscious rat. The device (asterisk) infuses fluid and senses pressure in the eye via a detachable tether that connects to a pin-port (arrowhead) mounted atop the head. (C) Schematic of system components. The device consists of a micropump, microcontroller, flow restrictor, pressure transducer, and connector tubing. The device is powered by and communicates with a laptop computer via a USB connector. Fluid is drawn from the saline reservoir and infused into the anterior chamber of the eye though a cannula.

level by modulating flow rate in proportion to the setpoint error (CP mode). A custom Labview® (National Instruments, Austin, TX, USA) program running on a laptop communicates with the system and stores recorded pressure and flow data.

Fig 2 provides a lumped parameter model of the eye and perfusion system. The eye model is a dynamic version of the Goldmann equation [18],

$$IOP = R_T(F_A - F_U) - EVP \qquad [1]$$

in which the time-varying IOP ($P_E$) is determined by the balance of fluid volume flowing in from aqueous humor production ($F_A$), flowing out through IOP-independent uveoscleral pathways ($F_U$) and through trabecular pathways of resistance ($R_T$) into episcleral veins of pressure ($P_V$), and filling the globe of compliance ($C_G$). The perfusion system is modeled by a pump of variable flow rate ($F_P$), flow restrictor of resistance ($R_S$), connector tubing of compliance ($C_S$), and cannula of resistance ($R_C$). Tubing resistance is very small relative to the fine diameter cannula and included in $R_C$. Also depicted are two sensors that are used only for system calibration and testing: an identical pressure sensor to measure pump pressure head ($P_P$) and an inline flow sensor (LG16-015, Sensirion Inc, Chicago, IL) to measure flow through the cannula ($F_C$). Per manufacturer specifications $P_P$ and $F_P$ both depend on pump duty cycle,

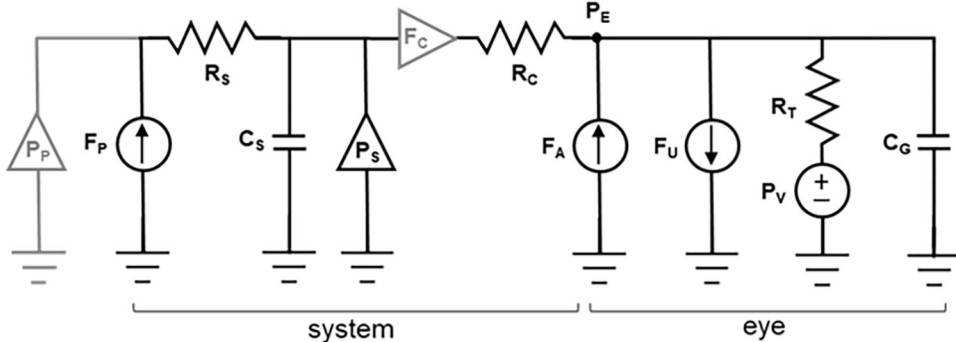

**Fig 2. System design and calibration.** (A) Equivalent circuit model of the eye and eye perfusion system components. Gray elements represent pressure and flow sensors used for device calibration and testing only and are not part of the fabricated device. $P_P$: pump pressure head, $F_P$: pump infusion rate, $R_S$: flow restrictor resistance, $C_S$: connector tubing compliance, $P_S$: sensor pressure, $F_C$: flow through cannula, $R_C$: cannula resistance, $P_E$: intraocular pressure, $F_A$: aqueous production rate, $F_U$: uveoscleral outflow, $R_T$: trabecular outflow resistance, $P_V$: episcleral venous pressure, $C_G$: globe wall compliance.

which is the percentage of time the pump is on during a power cycle, and pump output impedance. The multifactorial relationship complicates system operation and was simplified by using a high-impedance flow restrictor to turn the pump into a constant pressure source. S1 Appendix is provided to help clarify pump control and how the dependence on output impedance is eliminated at high $R_S$.

## Benchtop testing

The hydrodynamic properties and performance of fabricated eye perfusion systems were evaluated with benchtop tests. Prior to testing, it was crucial to eliminate any bubbles in the system for consistent and proper operation. This was done by carefully filling each component as the system was assembled and running the pump for at least 2 hours, after which tubing lines were inspected to verify bubble clearance. Pressure sensors were then calibrated by mercury manometry to have correct gain and offset, and the commercial calibration of flow meters was checked. $C_S$ was determined by replacing the pump with a 5-µl glass syringe (Hamilton Company, Reno, NV, USA), injecting a small bolus of fluid of increasing volume with the cannula sealed with cyanoacrylate, and linearly regressing bolus volume against measured $P_S$. $R_S$ and $R_C$ were determined and their stability assessed by repeatedly incrementing pump duty cycle in 2% steps from 0–100% round-the-clock for two months. $P_P$, $P_S$, and $F_C$ were measured for each step with the cannula immersed in saline at the same height as the system to eliminate hydrostatic pressure or surface tension effects on fluid flow. $R_S$ and $R_C$ were respectively tracked over time by linearly regressing measured pressure drops across each element against measured flows. $R_C$ was determined for two different cannulas: a 33-gauge needle (TSK Laboratory, Tochigi, Japan) and a silicone tube (outer and inner diameter: 200 and 100 µm, As One International, Santa Clara, CA, USA). Once $C_P$, $R_S$, $R_C$, and the relationship between $P_P$ and duty cycle were known, a given system could be programmed to output any desired $F_P$ from measurements of $P_S$. Pump rate was incremented in 5 µl·min$^{-1}$ steps to confirm that $F_P$ equals measured $F_C$ for all steps. System accuracy was assessed by inserting the needle cannula into a polyamide microtube (inner diameter: 63.5 µm, length: 10 cm, Microlumen Inc, Oldsmar, FL, USA) of similar resistance as rat eyes, immersing in saline, and measuring $P_S$ for step increases in $F_P$. Pressure-flow data were linearly regressed to estimate microtube resistance, which was compared with the expected resistance based on Poiseuille's law.

## Animal preparation

Animal experiments were conducted in accordance with the ARVO Statement for the Use of Animals in Ophthalmic and Vision Research and in compliance with a protocol approved by the Institutional Animal Care and Use Committee at the University of South Florida. Adult Brown-Norway rats (retired breeding males, 300–400 g) were purchased from a commercial vendor (Charles River Laboratories, Wilmington, MA) and housed in a 12hr light/12hr dark cycle with food and water ad libitum.

## Anesthetized animal experiments

Animals were anesthetized with an intraperitoneal injection of ketamine hydrochloride (75 mg·kg$^{-1}$) and xylazine (7.5 mg·kg$^{-1}$), supplemented as needed until the femoral vein was cannulated. Anesthesia was thereafter maintained by intravenous ketamine infusion at a rate (30 mg·kg$^{-1}$·hr$^{-1}$) that was adjusted as needed to keep heart rate at physiological levels. Body temperature was monitored with a rectal thermometer and kept at 37˚ by a regulated heating pad. The head was secured in a stereotactic unit, the system was connected to a 33-gauge needle, and the needle was advanced into the anterior chamber of one eye using a micromanipulator with care taken to avoid damaging the iris or lens. The cornea was hydrated with a constant saline drip and intermittently checked to ensure there was no needle movement or fluid leakage, especially at higher pressure levels. A bolus of fluid was injected to inflate the eye after cannulation, and resting IOP was defined as the level at which linear regression of $P_E$ data exhibited no statistically significant slope over a 3-min period. For CF experiments, the system was configured to increment $F_P$ in 0.2 μl·min$^{-1}$ steps to a maximum rate of 1 μl·min$^{-1}$ while recording $P_E$ every second. Each step lasted until IOP reached a new steady-state level, as defined by no significant slope in $P_E$ data over the prior 3-min period. For CP experiments, the system was configured to increment $P_E$ in 5 mmHg steps from resting IOP up to 50 mmHg. The system dynamically adjusts $F_S$ to accomplish this, so each step lasted until pressure and flow data both exhibited no significant slope over the prior 3-min period. The accuracy of $F_P$ data was validated by simultaneously recording $F_C$ with a flowmeter. In some cases, the accuracy of $P_E$ data was validated by inserting a second needle in the eye and directly recording IOP with another pressure sensor. Pressure-flow data were linearly regressed to estimate $R_T$, the reciprocal of which is conventional outflow facility.

## Awake animal experiments

Animals were temporarily anesthetized with the ketamine-xylazine mix, and the tip of a silicone cannula was surgically implanted in the anterior chamber of one eye. Details of the cannula implantation procedure have been published [19, 20]. In short, the cannula was filled with balanced salt solution, and the tip was backloaded with a mixture of 25% moxifloxacan (Vigamox®, Cardinal Health, Dublin, OH) and 1% enoxaparin (Lovenox®, Covetrus, Portland, ME) in saline to mitigate inflammatory responses that can clog the tip [7]. The cannula was connected to a custom head mount affixed to the skull with bone screws and dental cement, passed subdermally to the orbit, and secured with 11–0 scleral sutures in a z-pattern to prevent the tip from slipping out of the eye. The cannula tip was inserted just posterior to the circumlimbal vessels and entered either immediately anterior or posterior to the iris. After at least three days of recovery, the system was placed next to the animal's cage and connected with 25-gauge PTFE tubing (length: 45 cm) that ran inside a metal spring-like cable and swivel to a PinPort™ (Instech Laboratories, Plymouth Meeting, PA) in the head mount (Fig 1B). The tether setup protected tubing from animal bites and prevented tether knotting, respectively, and the pinport allowed for rapid system attachment and detachment with minimal impact on

IOP, which was useful for animal handling and tubing repairs if necessary. The system was inspected regularly for bubbles or leaks and for signs of eye injury or cannula clogging.

Outflow facility measurements in conscious animals required modification of the CP method to combat heightened levels of IOP noise. The modification involved formulating real-time "noise free" estimates of IOP ($\tilde{P}$) and flow ($\tilde{F}$) responses to pressure steps based on the eye model in Fig 2A. Estimates were updated each second by regressing incoming raw data repeatedly by the model-derived fitting functions $\tilde{P}(t) = A_P(1 - e^{-t/\tau}) + B_P$ and $\tilde{F}(t) = A_F e^{-t/\tau} + B_F$, where $A$ and $B$ are arbitrary scaling and offset parameters and $\tau$ is response time constant [18]. The recursive regression algorithm commenced 2 mins after step onset and continued until steady state was reached, which was defined empirically as $\tilde{P}$ deviating less than 0.5 mmHg and $\tilde{F}$ less than 0.05 µl·min$^{-1}$ over a 2-min period that is more than 6·$\tau$ after step onset. The algorithm is able to converge because it grows less sensitive to transient IOP fluctuations with time, as more data get incorporated into the regression. Outflow facility was measured every 2 or 6 hrs for several days by raising IOP in 5-mmHg steps to a maximum of 40 mmHg. The different intervals traded off temporal resolution with pressure perturbation frequency. Resting IOP was checked by tonometry (TonoVet, iCare, Raleigh, NC, USA) with the animal under brief isoflurane anesthesia.

## Data analysis

Data were analyzed using SigmaPlot software (Systat, Inc., San Jose, CA, USA) with statistical significance assessed at an α-level of 0.05 using two-tailed t-tests. All data passed a Shapiro-Wilk normality test, and all paired data also passed a Brown-Forsythe equivariance test. Results are expressed as mean ± standard deviation. Data correlation was assessed with the adjusted $R^2$. Linear regressions for steady-state detection in anesthetized animals were performed in Labview with statistical significance of the slope assessed at an α-level of 0.001. Facility data from conscious animals were considered outliers and excluded from statistical analysis if below 0 or above 60 nl·min$^{-1}$·mmHg$^{-1}$, as the former is physiologically impossible and the latter would be well outside the reported range for Brown Norway rats [15, 21].

## Results

### System characterization and validation

A total of 5 portable eye perfusion systems were fabricated, and their hydrodynamical properties were characterized in terms of four parameters: $F_P$, $R_S$, $C_S$, and $R_C$. $F_P$ depends on pump duty cycle and pump output impedance. It cannot be explicitly specified but can be deduced from the pressure drop measured across the flow restrictor if $R_S$ is known. Fig 3A plots repeated measurements of $R_S$ over several weeks for one system. It can be seen that $R_S$ held steady throughout the experiment, indicating the flow restrictor did not become leaky or occluded (slope = -0.019 ± 0.025 mmHg·min·µl$^{-1}$·day$^{-1}$, p = 0.16, n = 5). $R_S$ averaged 134 ± 51 mmHg·min·µl$^{-1}$ across systems (n = 5). As shown in S1 Appendix, the high output impedance causes the pump to behave like a constant pressure source. Fig 3B plots repeated measurements of pump pressure head as a function of duty cycle. $P_P$ exhibited a sigmoidal dependence that was stable over several weeks of testing. The non-saturated region from 0 to ~75% duty cycle was well fit by a third-order polynomial ($R^2$ = 0.999, n = 5), the parameters of which were used to map duty cycle to a corresponding $P_P$ for each system. $F_P$ is then calculated from $P_P$, $R_S$, and measured $P_S$. Fig 3C compares computed $F_P$ with measured $F_C$ for all systems. The two should be equal since all pump outflow passes through the flowmeter in steady state. The zero offset and unity slope of the regression line ($R^2$ = 0.999, n = 5) demonstrates that $F_P$ is accurate and can be set at rates up to about 2 µl/min. System compliance was measured by

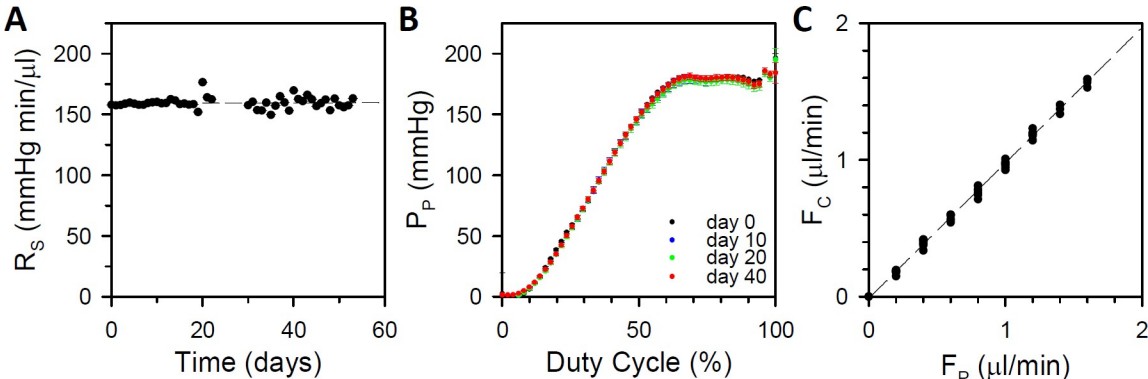

**Fig 3. Bench testing.** (A) Flow restrictor resistance measured over several weeks of continuous saline effusion by a fabricated system. Dashed line is a linear fit of the data ($R_S$ = 159 + 0.010·days). (B) Pump input-output curve measured on different days over a several week period. (C) Measured flow rate ($F_C$) for 5 fabricated systems across a range of pump rate settings ($F_P$). Dashed line is a linear fit of the data ($F_C$ = -0.01 + 0.99·$F_P$).

sealing the cannula and injecting a fluid bolus of increasing volume. $C_S$ averaged 0.04 ± 0.01 mmHg·$\mu$l$^{-1}$ across systems (n = 5). Cannula resistance was calculated from the measured pressure drop produced by $F_P$ with the tip immersed in saline. $R_C$ averaged 1.0 ± 0.1 mmHg·min·$\mu$l$^{-1}$ for needle cannulas and 2.2 ± 0.1 mmHg·min·$\mu$l$^{-1}$ for silicone cannulas (n = 5). After system characterization, performance was validated with a load of known eye-like conductance (36 nl·min$^{-1}$·mmHg$^{-1}$) while recording flow concurrently with a flowmeter. Conductance estimates with the system (35 ± 2 nl·min$^{-1}$·mmHg$^{-1}$, n = 5) were not significantly different from those with the flowmeter (36 ± 2 nl·min$^{-1}$·mmHg$^{-1}$; p = 0.63) or from the known load conductance (p = 0.29).

## Acute measurements of ocular fluid dynamics

The system was next tested on anesthetized rats. Since $P_E$ is computed by the system from measured $P_S$, it was checked by directly recording IOP with a second needle in the eye. $F_P$ was also validated with a flowmeter. Fig 4A plots pressure-flow data from a CF and CP experiment. $P_E$ and $F_P$ closely tracked measured IOP and $F_C$ in both experiments, except perhaps at CP step onset where sharp transients in $P_E$ and $F_P$ were not recorded. Fig 4B and 4C show that response waveforms were highly correlated over a wide range of pressures ($R^2$ = 1.00) and flows ($R^2$ = 0.88). The regression slope was near unity across animals (1.01 ± 0.03, n = 5), demonstrating that system output is accurate for living eyes.

Fig 5A plots $P_E$, $F_P$, and $F_C$ data for a series of CF and CP steps. It can be seen that IOP settling time is much faster in CP mode since fluid is infused at high rate at step onset and promptly lowered to the steady-state rate as $P_E$ nears the CP setpoint. Pressure-flow data were collected in under 30 mins as compared to over 2 hrs in CF mode. It can also be seen that $F_P$ and $F_C$ did not always track together at step onset. This is because system compliance must first get filled for fluid to enter the eye. Fig 5B shows that computed and measured flow data stepped to similar levels in both experiments, yielding nearly identical estimates of outflow facility. Fig 5C and 5D summarize pressure-flow data for all CP experiments (n = 11), showing the relatively linear nature of the results. Facility estimates with the system (23 ± 3 nl·min$^{-1}$·mmHg$^{-1}$) were not significantly different from those with the flowmeter (22 ± 4 nl·min$^{-1}$·mmHg$^{-1}$, p = 0.99) or with reported values for Brown-Norway rats (p = 0.63) [15, 21]. Having validated system performance on anesthetized animals, $P_E$ reported by the system is henceforth considered IOP.

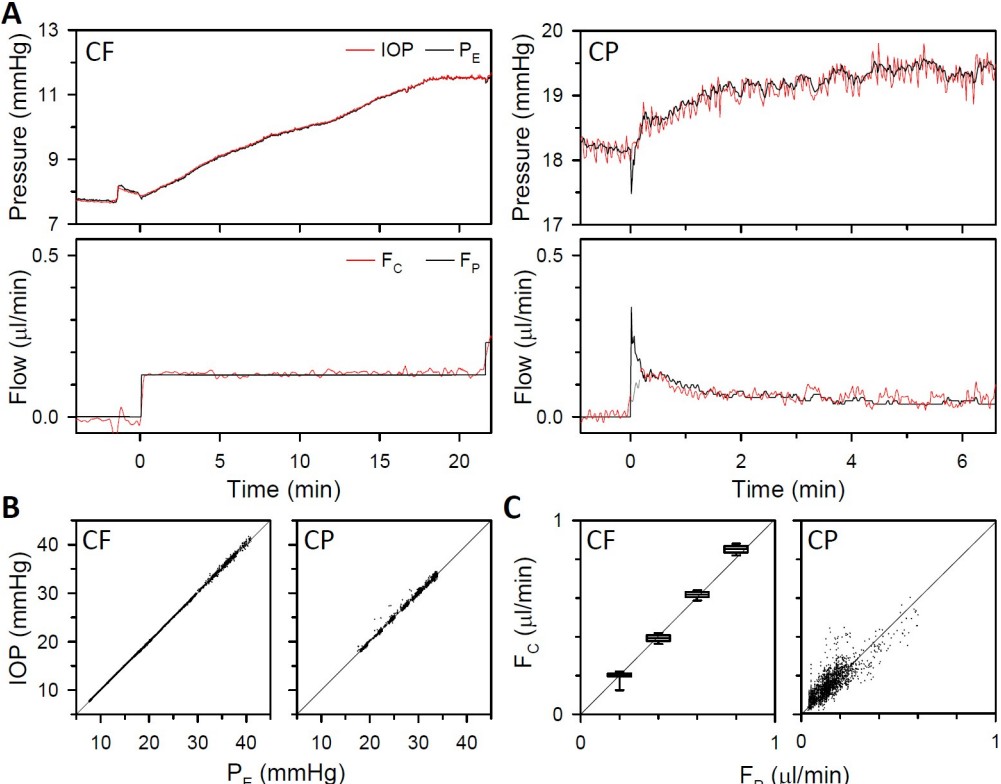

**Fig 4. System performance in anesthetized animals.** (A) Left, pressure (top) and flow (bottom) records from a ketamine-anesthetized rat in response to a single pump-driven step increase in flow (CF). Right, pressure and flow records from another rat in response to a single pump-driven step increase in pressure (CP). Black and red traces correspond to eye pressure and flow reported by the system ($P_E$ and $F_P$) and independently measured by pressure and flow sensors (IOP and $F_C$), respectively. (B) Comparison of system and sensor pressure readings across a series of CF (left) and CP (right) steps. Solid lines are linear regression fits (slope: 1.002 and 1.001) (C) Comparison of system and sensor flow readings across a series of CF (left) and CP (right) steps. A box-and-whisker plot is used for the former since flow is fixed for each step. Solid lines are linear regression fits (slope: 1.006 and 0.993).

## Chronic measurements of ocular fluid dynamics

Lastly the system was deployed in awake free-moving rats. Of the 7 animals implanted with cannulas and tethered to the device, 4 animals yielded facility data for one or more days. Fig 6A plots IOP and flow data recorded over a 5-hr period from a rat during which facility was measured at 8:00, 10:00, and 12:00. It can be seen that IOP is highly variable, sporadically fluctuating by several mmHg over fast and slow time scales due to assorted internal and external factors [7, 22, 23]. Some of the variability may also be attributed to head and body motion artifacts [5, 24, 25]. The fluctuations complicate estimation of steady-state responses to step changes in pressure or flow and can destabilize feedback control of the pump. To surmount these challenges, IOP and flow data were processed in real time by a recursive regression algorithm based on the eye model in Fig 2. Fig 6B shows raw and processed records for the 8:00 facility measurement. The processed record is smoother because the algorithm uses the eye model to estimate what IOP and flow would be in absence of noise based on response trajectories. Fig 6C plots processed pressure-flow data for each step series. Facility estimates were fairly reproducible despite the heightened IOP noise in conscious animals (23, 27, and 28 nl·min$^{-1}$·mmHg$^{-1}$ for 8:00, 10:00, and 12:00 respectively).

Fig 7A plots IOP and flow data recorded over a 24-hr period from a different animal in which outflow facility was measured at 3:00, 9:00, 15:00, and 21:00. A diurnal rhythm in resting

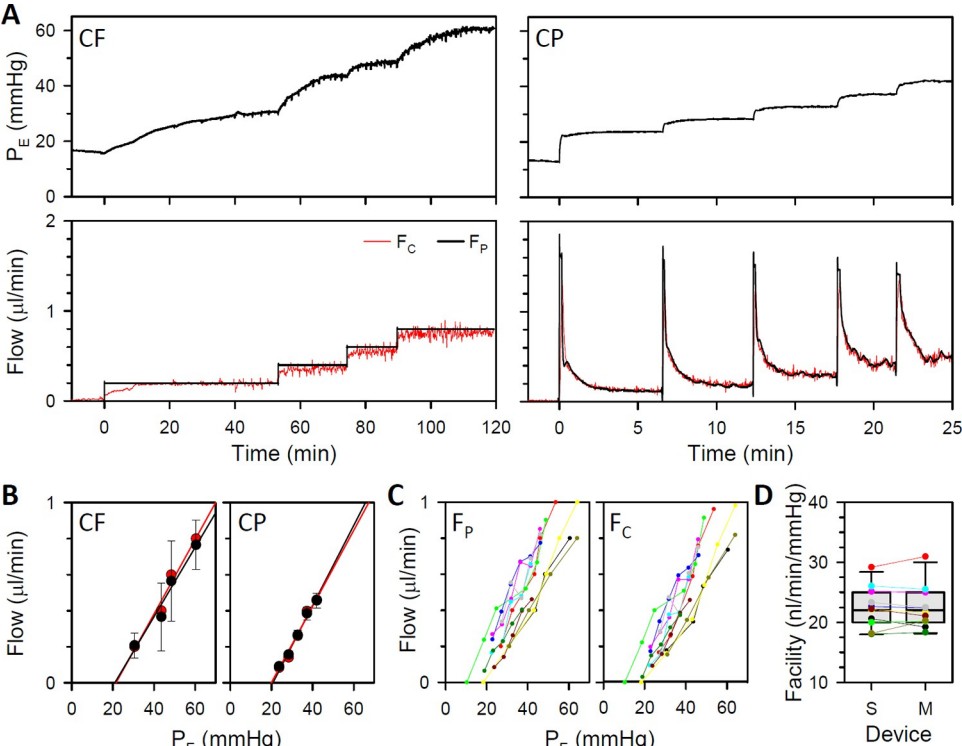

**Fig 5. Outflow facility measurements in anesthetized animals.** (A) Records of pressure (top) and flow (bottom) from a ketamine-anesthetized rat in response to a series of flow steps (CF, left) and from another rat to series of pressure steps (CP, right). Black traces correspond to eye pressure and flow reported by the system ($P_E$ and $F_P$) and red traces to flow measured independently by a flow meter ($F_C$). (B) Steady-state pressure-flow data for the CF (left) and CP (right) experiment. Black and red lines are linear fits of the data (CF intercept: -0.41 and -0.44 $\mu l \cdot min^{-1}$, CF facility: 19 and 21 $nl \cdot min^{-1} \cdot mmHg^{-1}$, CP intercept: -0.45 and -0.42 $\mu l \cdot min^{-1}$, CP facility: 22 and 21 $nl \cdot min^{-1} \cdot mmHg^{-1}$). Error bars give standard deviation. (C) Pressure-flow data from 11 anesthetized animals (colored symbols) with the system (left) and flow meter (right) (D) Box-and-whisker plot of facility estimates with the system (S) and flow meter (M) for the same animals (colored symbols). Y-intercepts respectively averaged -0.36 ± 0.12 and -0.41 ± 0.14 $\mu l \cdot min^{-1}$ (p = 0.44).

IOP is apparent, swinging from ~10 mmHg during the day to ~25 mmHg during the night. Also apparent is a marked day-night difference in IOP and flow responses to virtually the same step series. Fig 7B and 7C plot processed pressure-flow data for each step series. In addition to a diurnal shift in x-intercept, reflecting the swing in mean IOP, a diurnal change in slope can be seen. Facility estimates are highest during the day (9:00 and 15:00, 32 and 36 $nl \cdot min^{-1} \cdot mmHg^{-1}$) when IOP is low and lowest at night (3:00 and 21:00, 23 and 22 $nl \cdot min^{-1} \cdot mmHg^{-1}$) when IOP is high. Fig 8 summarizes results from three other animals in which outflow facility was measured periodically for multiple days. A diurnal rhythm in facility was observed in each experiment, with daytime (6AM– 6PM) estimates significantly higher than nighttime (6PM– 6AM) estimates (Y45 [2 days]: 18 ± 6 $nl \cdot min^{-1} \cdot mmHg^{-1}$, n = 14, v. 11 ± 3 $nl \cdot min^{-1} \cdot mmHg^{-1}$, n = 14, p < 0.001; Y72 [7 days]: 24 ± 7 $nl \cdot min^{-1} \cdot mmHg^{-1}$, n = 48, v. 17 ± 7 $nl \cdot min^{-1} \cdot mmHg^{-1}$, n = 42, p < 0.001; Y77 [4 days]: 25 ± 14 $nl \cdot min^{-1} \cdot mmHg^{-1}$, n = 10, v. 12 ± 6 $nl \cdot min^{-1} \cdot mmHg^{-1}$, n = 7, p = 0.03).

## Discussion

We have presented a small portable device capable of autonomously manipulating IOP via a cannula implanted in the eye. The device is designed for tethered use on rats and could be

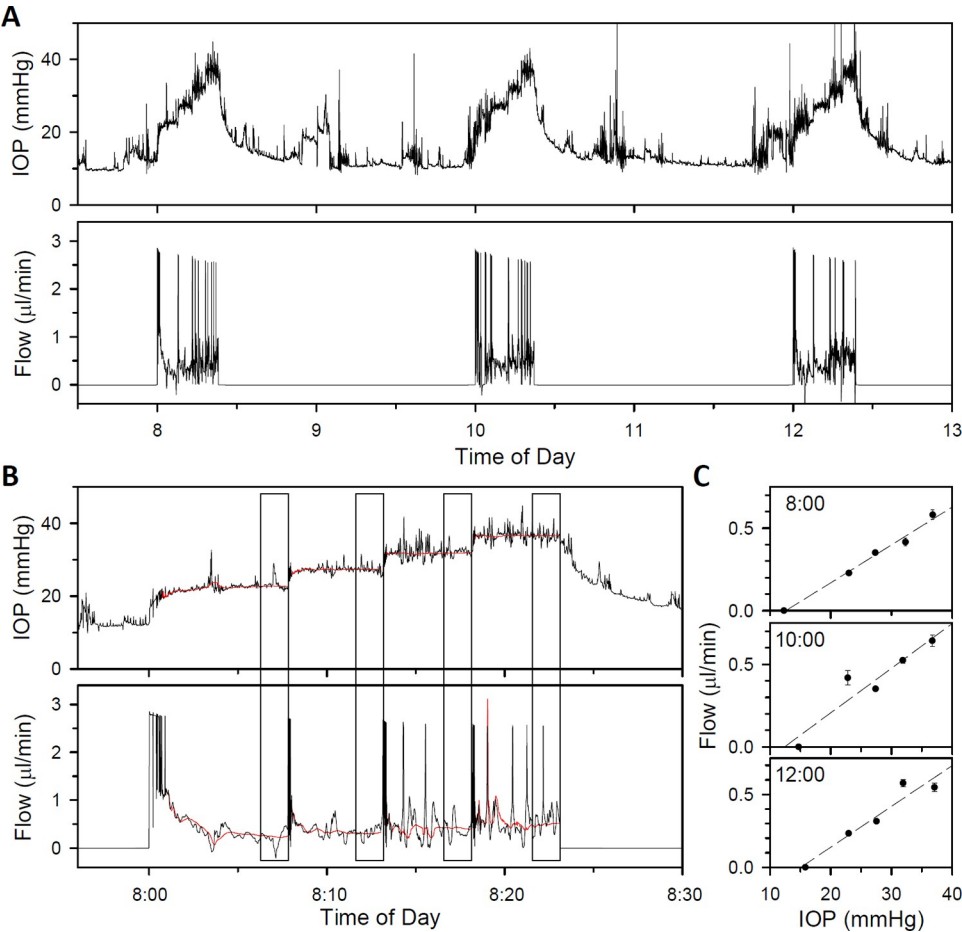

**Fig 6. System performance in conscious animals.** (A) Records of pressure (top) and flow (bottom) from an awake free-moving rat over a several hour period. Outflow facility measurements were made at 8A, 10A, and 12P with the system in CP mode. (B) Pressure and flow records for the facility measurement at 8A. Black and red traces respectively plot the raw data and the data after real-time processing with a recursive regression algorithm which is designed to ignore transient IOP fluctuations that can interfere with steady-state estimation. Boxes indicate the 2-min window in which the processed signal met steady-state criteria, triggering the system to initiate the next pressure step or end the series. (C) Steady-state pressure-flow data from processed records of the 3 facility measurements. Dashed lines are linear fits of 8A, 10A, and 12P data (intercepts: -0.29, -0.34, and -0.42 $\mu$l·min$^{-1}$, facility: 23, 27, and 28 nl·min$^{-1}$·mmHg$^{-1}$). Error bars give standard deviation.

readily modified for the fluid volumes and flow rates of larger eyes. One key component of the design is a high-impedance flow restrictor that causes a piezoelectric pump fabricated as a controllable flow source to behave like a constant pressure source (S1 Appendix). The restrictor renders the pump insensitive to loading by the cannula and eye and to variation in these loads over time or between animals. It also allows for infusion of fluid at microfluidic rates and for specification of flow rate from pressure measurements. The absence of a flowmeter greatly reduces device size and cost and avoids additional pump loading, which can markedly dampen system dynamics if the flowmeter conduit is narrow [18]. Device testing showed that flows can be delivered at rates comparable to aqueous humor production, that pressure generation and restrictor impedance do not vary over more than a month of continuous operation, and that the fluidic resistance of a known load can be accurately measured. A second key element is a modified proportional feedback algorithm that continually adjusts pump operation in the face of physiological noise to minimize data collection time without loss of accuracy. The novel

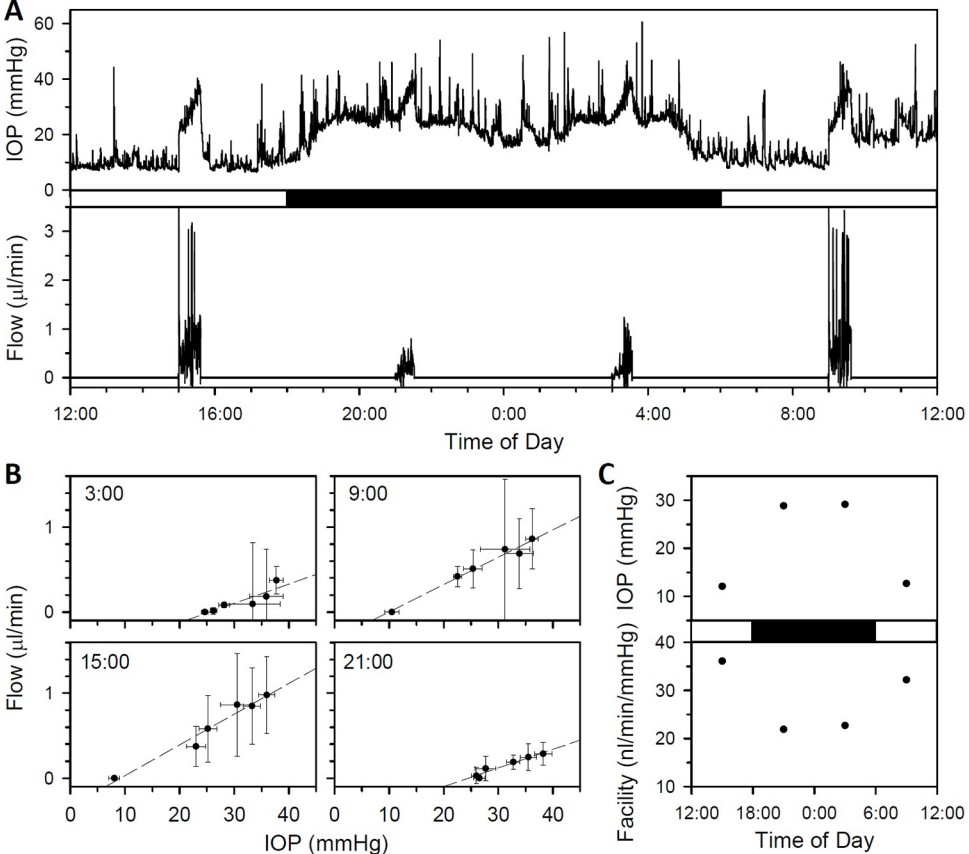

**Fig 7. Outflow facility measurements in conscious animals.** (A) Records of pressure (top) and flow (bottom) from an awake free-moving rat over a 24-hr period. Outflow facility measurements were made at 3P, 9P, 3A, and 9A with the system in CP mode. (B) Steady-state pressure-flow data from processed records of the 4 facility measurements. Dashed lines are linear fits of 3A, 9P, 3A, and 9A data (intercepts: -0.58, -0.32, -0.33, and -0.54 $\mu$l·min$^{-1}$, facility: 23, 32, 36, and 22 nl·min$^{-1}$·mmHg$^{-1}$). Error bars give standard deviation. (C) Comparison of IOP and facility estimates throughout the day. White and black boxes in A and C indicate the light and dark phases of the ambient lighting cycle.

algorithm does this by applying a viscoelastic model of the eye to formulate a real-time feedback estimate of pressure and flow. Device assessment in anesthetized and conscious rats showed that outflow facility can be accurately measured in much less time than standard CF and CP techniques, which get unsettled by transient IOP fluctuations. The device is thus uniquely capable of monitoring facility round-the-clock in awake free-moving animals as well.

Initial experiments with the device in conscious rats have revealed a diurnal rhythm in conventional outflow facility. This is the first demonstration of a facility rhythm in rodents. Such a rhythm has been reported in rabbits and humans using fluorophotometry [26, 27]. Facility was highest during the day and lowest at night, swinging 9 ± 4 nl·min$^{-1}$·mmHg$^{-1}$ (39 ± 17% of the daytime level) on average across tested animals. The cyclic swing would predict an IOP change of 8 mmHg based on the aqueous production rate of 0.35 $\mu$l·min$^{-1}$ reported for rats [28], if uveoscleral outflow is neglected, which is consistent with the known circadian rhythm in rat IOP [24, 29–31]. It might not be the only contributor to the IOP rhythm as diurnal rhythms in aqueous production and uveoscleral outflow have been documented in rabbits and humans as well [26, 27]. The facility results cannot be easily explained by variations in aqueous production or uveoscleral outflow either. $F_A$ and $F_U$ would have to change systematically with pump-induced increases in IOP so as to conflate with a facility change, but they are considered

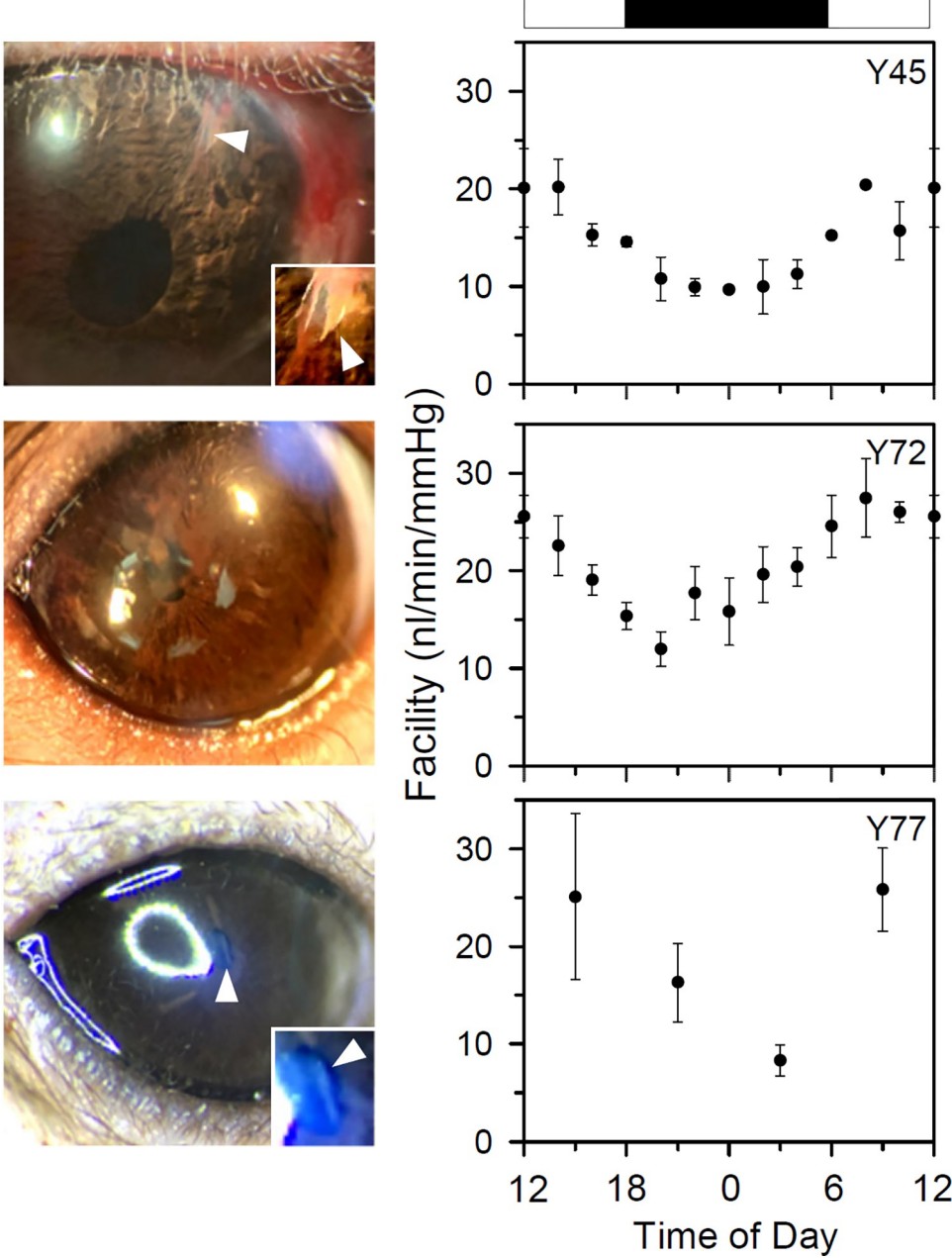

**Fig 8. Diurnal rhythms in outflow facility.** Summary of round-the-clock facility estimates across multiple days in three additional conscious rats (Y45: 2 days, Y72: 7 days, Y77: 4 days). Error bars give standard error. White and black boxes indicate the light and dark phases of the ambient lighting cycle. Images show eyes on final day of experiment. Arrow in Y45 and Y77 points to cannula, and insets in lower right are a close-up of the cannula tip. Inset contrast was adjusted to enhance tip visibility. Cannula tip was beneath the iris and cannot be seen in Y72.

pressure independent. The Goldmann equation would require revision if this were not the case. Also, the timescale of diurnal aqueous production and uveoscleral outflow rhythms is much too slow to have significantly altered outflow over the half-hour duration of each facility measurement. Further evidence that they are likely constant is the subsequent return of IOP to its resting level in Figs 6A and 7A. Interestingly, oscillations in outflow facility were not

apparent in multi-day measurements from enucleated eyes [32], implying that circadian rhythmicity is not an intrinsic property of anterior segment tissues. The eye presumably must remain connected to circulatory and neural signals from the suprachiasmatic nucleus, which has been shown to regulate IOP rhythmicity [33, 34]. Observed changes in outflow resistance may reflect sympathetic shrinkage of Schlemm's canal [35], suppression of trabecular meshwork phagocytosis [36], or other processes under circadian control.

Outflow facility estimates with the system are consistent with previous measurements on anesthetized rats. Feola et al. [21] reported values of 21.2 [11.2, 40.0] and 26.1 [23.8, 28.7] nl·min$^{-1}$·mmHg$^{-1}$ for young and middle-aged Brown Norway rats, respectively, while Ficarrotta et al. [15] reported a value of 23 ± 2 nl·min$^{-1}$·mmHg$^{-1}$ for retired-breeder Brown Norway rats. Facility values in this study (23 ± 4 nl·min$^{-1}$·mmHg$^{-1}$) are within margin of error of their findings, and all are around two-fold less than reported values for albino rat species [37, 38]. Moreover, repeated estimates of outflow facility in conscious rats exhibited similar variability as cited studies performed on different rats [15, 21]. This not only offers support for device performance in free-moving animals, but also suggests that some of the variability in anesthetized rat experiments might be attributed to circadian rhythmicity. Systematic changes in facility may be noted during daytime hours when such experiments are typically performed.

## Applications for ocular research

The device opens several avenues of study by virtue of its capacity to monitor and manipulate IOP in awake free-moving animals. Firstly, aqueous humor dynamics can be directly measured free of any anesthetic effects. Prior work has examined the impact of varying levels of anesthesia on IOP and facility [17], but animal sedation was still necessary due to limitations of available eye perfusion technology. Secondly, acute or chronic ocular hypertension can be induced in animal models via feedback control of IOP rather than by experimentally damaging outflow pathways. Constant IOP elevation via a gravity-fed perfusion system was shown to cause graded optic nerve degeneration and retinal ganglion cell loss [20]. Pathophysiological responses of an otherwise normal functioning eye can now be examined with reproducible insults having any desired IOP profile. Thirdly, the pin-port design allows for tether detachment and delivery of putative therapeutic drugs through the implanted cannula (dead volume: 3–5 μl) directly into the anterior chamber of animals with experimental glaucoma, reducing the quantity of drug necessary and the delay in arrival at target tissue compared to extraocular delivery. Medicinal effects can also be evaluated in relation to changes in IOP mean and variance over long periods of time. And fourthly, outflow facility and other bulk fluid dynamic properties of the eye can be chronically measured in healthy and glaucomatous animals. Automated round-the-clock monitoring is particularly valuable for circadian research since quantifying rhythmic changes in these variables via tonometry or fluorophotometry would be laborious and only a few readings could be made per day.

## Comparison to existing technology and device limitations

The principal advantage of the device is that no commercial technology available at present can measure aqueous humor dynamics in conscious animals. An earlier design of this portable smart pump was introduced that can also manipulate IOP [19]. It employs a different feedback algorithm that infuses fluid at fixed rate and turns the rate on and off so as to keep IOP within a narrow range of a user-specified level. A major upgrade to the current design is the acquisition and feedback control of flow based on recorded IOP. The smooth and continuous adjustment of pump rate speeds convergence to the IOP set point and eliminates oscillatory flow patterns. Another is the implementation of a modified CP algorithm to overcome IOP

variability that was encountered in conscious animals. Both devices can also be used in anesthetized animals at a fraction of the cost and space of a comparable laboratory setup. A typical CF perfusion system of a programmable syringe pump, pressure sensor, and cables or CP system of an adjustable-height saline reservoir, pressure sensor, flowmeter, and connective tubing would be replaced by a palm-size box. Unlike benchtop setups, the device would though need modification to generate flow rates suitable for larger eyes than rats. Rates up to 40 $\mu l \cdot min^{-1}$ could be produced by lowering restrictor resistance to 5 $mmHg \cdot min \cdot \mu l^{-1}$ before the system would become susceptible to external loading. The device would also require more maintenance to prevent buildup that may clog the flow restrictor with intermittent use and frequent calibration checks to ensure proper functioning.

A limitation of all CP systems is noise susceptibility. IOP noise perturbs gravity- and pump-driven control of flow. The perturbations are manageable in anesthetized animals, allowing for rapid and reliable estimation of steady-state pressures and flows. However, they are significant in awake free-moving animals. The feedback sensitivity of the device, like other pump-driven CP systems [13, 14], is set programmatically. If set too high, the device can exhibit unstable behavior due to mismatch in timing and magnitude of flow adjustments with incoming noisy pressure data. There is no risk of instability with gravity-fed systems because feedback control is instantaneous. The device employs a modified CP algorithm to smooth the feedback signal and optimize performance in conscious rats. Nevertheless, IOP variability will impact the duration of pressure-flow data collection and limit the frequency of facility measurement. Monitoring may need to be staggered in time on successive days for circadian studies in order to adequately sample the diurnal cycle and minimize risk of glaucomatous damage from excessive exposure to elevated IOP.

Another limitation of the system is the surgical implantation of a cannula and ocular response to the foreign body. Experimental success and longevity require the eye to remain healthy and the device to remain functional. Some surgical experience and good technique are needed to minimize tissue damage, prevent cannula extrusion, and avoid hyphema and other implantation issues. More challenging are inflammatory responses, which can lead to fibrin clots forming around the cannula and occlusion of the tip. In rats, the cannula is ultrafine so even a low-grade reaction can cause clogging. Leaving the eye undisturbed and using antifibrotics and steroids during the healing process can help dampen this response. We were able to obtain a ~60% success rate in 7 animals despite repeatedly infusing fluid into the eye at variable rates for one or more days. It is anticipated that ongoing efforts will continue to improve experiment success and longevity.

## Conclusion

In this paper, we have presented a new technology that provides for round-the-clock monitoring of aqueous humor dynamics in awake free-moving rats over long periods of time. The device generates microfluidic flow rates with similar precision as commercial syringe pumps, while allowing for automated measurement of IOP, outflow facility, and other biofluidic parameters free from any ocular effects of anesthesia. The device is small, inexpensive, and portable, offering new opportunities for glaucoma research and drug delivery.

## Supporting information

**S1 Appendix. Pump operation as a constant pressure source.**
(DOCX)

**S2 Appendix. Raw data of pump characteristics, pressure and flow traces, steady-state pressure-flow levels, and outflow facility estimates in anesthetized and conscious rats.**
(ZIP)

## Acknowledgments

Authors thank Christina Nicou for helping improve the eye cannulation procedure and Andrew Kaiser for surgical assistance.

## Author Contributions

**Conceptualization:** Christopher L. Passaglia.

**Data curation:** Youssef Mohamed.

**Formal analysis:** Youssef Mohamed.

**Funding acquisition:** Christopher L. Passaglia.

**Methodology:** Youssef Mohamed.

**Project administration:** Christopher L. Passaglia.

**Software:** Youssef Mohamed.

**Supervision:** Christopher L. Passaglia.

**Validation:** Youssef Mohamed, Christopher L. Passaglia.

**Visualization:** Youssef Mohamed, Christopher L. Passaglia.

**Writing – original draft:** Youssef Mohamed.

**Writing – review & editing:** Youssef Mohamed, Christopher L. Passaglia.

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
