## [Decision Letter · Decision Letter 0]

9 Sep 2022

PONE-D-22-19808A portable feedback-controlled pump for monitoring eye outflow facility in conscious ratsPLOS ONE

Dear Dr. Passaglia,

Thank you for submitting your manuscript to PLOS ONE. After careful consideration, we feel that it has merit but does not fully meet PLOS ONE’s publication criteria as it currently stands. Therefore, we invite you to submit a revised version of the manuscript that addresses the points raised during the review process.

We look forward to receiving your revised manuscript.

Kind regards,

Ted S Acott, PhD

Academic Editor

PLOS ONE

Journal Requirements:

2. As part of your revision, please complete and submit a copy of the Full ARRIVE 2.0 Guidelines checklist, a document that aims to improve experimental reporting and reproducibility of animal studies for purposes of post-publication data analysis and reproducibility: https://arriveguidelines.org/sites/arrive/files/Author%20Checklist%20-%20Full.pdf (PDF). Please include your completed checklist as a Supporting Information file. Note that if your paper is accepted for publication, this checklist will be published as part of your article.

"I have read the journal's policy and the authors of this manuscript have the following competing interests: U.S. patents 9022968, 9314375, and 10758408."

Additional Editor Comments:

reviewers were generally favorable to the study, but both had several concerns that would greatly improve the manuscript - please carefully address all reviewer concerns

Reviewers' comments:

Reviewer's Responses to Questions

**Comments to the Author**

1. Is the manuscript technically sound, and do the data support the conclusions?

Reviewer #1: Yes

Reviewer #2: Partly

2. Has the statistical analysis been performed appropriately and rigorously? 

Reviewer #1: N/A

Reviewer #2: I Don't Know

3. Have the authors made all data underlying the findings in their manuscript fully available?

Reviewer #1: No

Reviewer #2: Yes

4. Is the manuscript presented in an intelligible fashion and written in standard English?

Reviewer #1: Yes

Reviewer #2: Yes

5. Review Comments to the Author

Reviewer #1: Review: A portable feedback-controlled pump for monitoring eye outflow facility in conscious rats

This paper describes a small, portable device for continuously measuring IOP and outflow facility in

conscious, moving rats. Most systems require anesthesia and do not continuously monitor IOP and

outflow facility over long time periods, so this device has great potential in expanding the type of ocular

studies we can do in animals, such as circadian studies of changes in outflow facility and facility

measurements without impact from anesthetic or other changes to physiology. Thus, it will garner

interest from readers, and I am generally positive about its publication.

Overall, this system could be really useful for researchers. The long-term measurements in awake,

moving animals shown in Figures 7 and 8 are really interesting, and show great potential for work

studying rhythmic changes in aqueous humor dynamics. It would be nice to see this work eventually

used in an established glaucoma or ocular hypertensive rat model to further validate the IOP and facility

measurement capabilities. Below are detailed comments.

My main concern with this paper is that the authors have not put their data in context. Others have

measured outflow facility in the rat eye, including the high-fidelity measurements of Feola et al. and

previous measurements by the current authors, plus possibly others that I am not aware of. These

should be compared with the data gathered here. Similarly, the flow-pressure graphs (Fig 5B) can be

used to get an estimate of aqueous inflow rate by extrapolating the lines back to (estimated) EVP, at

which point the (negative) pump flow rate should equal inflow rate. Obviously, there are assumptions

underlying such a calculation, but it would be good to see what values the authors get. Measuring inflow

rate in small eyes like the rat is very technically challenging and having more data points, with suitable

caveats, is useful. Finally, what is the success rate? The authors show 3 animals that seem to have

worked, but how many were tried? I know there was some information on this in previous papers, but

now that the authors have more experience, a quantitative update should be given.

Other comments:

The first two figures provide a good overview of the device size, setup, and use. The compact nature and

basic components of the device are clear from figure 1A, and it’s great that this is shown at the start of

the paper. I may have liked to see the device with the cannula attached to a conscious rat as well, but

this is partially addressed in figure 8. Request: please include a photo in Fig 1 of the device in use with a

rat.

One thing the authors do not comment on is how bubbles are avoided/dealt with. Bubbles are the bane

of all who perfuse eyes, and some comments on this important topic are needed.

The model shown in Figure 2A is useful to demonstrate Goldmann’s equation and visualize the device

setup in more detail. It also helps with interpretation of the figures throughout the paper, as readers can

reference how each variable in the future graphs relates to the rest of the system. However, I believe

there is an error in Fig 2A: the R_G element should not be there. I understand this is to represent

viscoelastic globe stretching, but that is not a fluid resistance element per se. Globe stretching can be

represented by the compliance C_G. In any case, R_G does not seem to be used anywhere in the paper,

so deleting it should not change anything so far as I can tell.

While the assessment of the pump/system parameters is important (Figs 2B-D), I have some concerns. It

would help if the dashed line were labeled as nominal load in Fig. 2B. More importantly, the slope of the

load line in Fig 2B (and the operating range in Figs 2C and D) are not in the right range. I understand that

this refers to the situation before the internal flow limiter is used, but Fig 2C somehow implies that the

load that the pump sees in operation is 0.1 mmHg min/ul, which is far from the resistance of the eye (by

1-2 orders of magnitude). Fig 2C is similarly misleading. For Fig 2D, we only care about the very bottom

part of the graph, right? The other lines are useless in practice, so a zoomed view of the bottom of the

graph (with more data) would be useful. This whole section was confusing and should be

reworked/better explained/changed.

Overall, the system validation seems sound (Fig 3), showing the repeatability of the device in a benchtop

setup before introducing the noise that results from measuring a live, conscious animal. A minor critique

is that in Figure 3B, the measurement points for different days are all on top of each other and mostly

indistinguishable, I wonder if colors would be better or if there is another way to visualize this data more

clearly.

Minor comment: when comparing the new system to the independently measured IOP values in

anesthetized animals (Fig. 4) and measuring IOP with the new system in conscious animal (Fig. 5), the

IOP as measured by the system is referred to as PE (to match the model in Fig. 2, which is helpful), but in

later figures looking at diurnal patterns of IOP in rats (Fig. 6-8), it is labeled as IOP. This should be made

consistent.

Additionally, it’s not clear what Figure 5C is adding. It provides information for some of the other

animals measured, but it is not easy to distinguish individual lines by the different shapes. It is not

explicitly stated that the shapes are different animals, and it’s not easy to see how many are shown. This

panel should be improved. Further, Fig. 5D would be better as a box and whisker plot rather than a bar

plot (although keeping the individual data points and connecting lines).

Figure 6 demonstrates the capabilities of the device, especially by showing how well the feedback

algorithm functions in a moving, awake animal in 6B. The text claims that the amount of variation shown

here in outflow facility across a few hours (23, 27, 28 nl/min*mmHg) is “fairly reproducible” (line 340),

but does not cite other studies that justify this claim – see also major comment above.

The data in figure 7 showing the differences in IOP and facility between night and day is really

interesting, and I think it’s a highlight of this paper. It shows how necessary this device is because there’s

so much more we can learn about diurnal changes in outflow using this device. However, it is confusing

that the setup of Fig. 7A is different from B and C (light/dark intervals are inconsistent). Please fix. Same

for the diurnal curves in Fig 8.

While I appreciate that Figure 8 shows the cannula located in different locations on the rat eye, and that

the diurnal patterns shown in 7 are still maintained in these other animals with the cannulas located at

different points, it is almost impossible to actually see the cannulas in the pictures of rat eyes in figure

8A. In the middle panel it is completely impossible – presumably some of the cannula was in the AC.

Zoomed in views/better labeling are needed.

The discussion mentions measuring ocular viscoelasticity (line 441), and although this is included in the

system model, data isn’t shown validating this measurement. Suggest just deleting this part.

Reviewer #2: This is a description of a new system for measuring aqueous humor dynamics in awake rats. The basic set up consists of a small silicone tube implanted in the anterior chamber that is connected externally to a “cage-side” apparatus capable of monitoring IOP and pumping balanced salt solution into the eye at microfluidic rates equivalent to aqueous humor formation or flow. These can be controlled through feedback from the measured IOP. For validation, the authors have used anesthetized animals to directly compare IOP and flow readings obtained by their system with independent measurements made through a secondary cannula placed in the eye. In implanted animals, they show the response of pressure and flow to paradigms of, respectively, constant flow and constant pressure--approaches that are commonly used in aqueous humor dynamics studies. Responses appear to be in line with what would be expected. The system is then used to obtain outflow facility readings in a small number of awake animals over time. They find evidence of a diurnal fluctuation in IOP equivalent to reports in the literature (with night-time IOP at about 25 and day-time of 15). Interestingly, they also report that conventional outflow facility also has a diurnal variation, being higher during the day and lower at night. Up to now such measurements have not been possible as existing systems require the use of either general anesthesia or the need for studying enucleated eyes.

Generally, this is a difficult manuscript to read. Much of this is due the use of a large number of acronyms, many of which do not appear intuitive, or are inconsistent. For example, CG stands for “globe wall compliance”, whereas C is generally used to indicate facility in some parts of this manuscript and generally in the aqueous humor dynamics literature. Also, line 97, they refer to “PF” mode. It’s not clear what this is, although it appears to be what they earlier define as “CP” (line 64). They persist in using this “PF” designation throughout the manuscript, without ever applying a definition. The authors need to carefully evaluate their acronyms, make sure they are consistent and use them as little as possible in order to minimize the amount of work they are asking of the reader. While a table providing definitions would be somewhat helpful, it would still be extremely cumbersome. A strong suggestion would be, in addition to a table, to spell out the terms in full in the body of the manuscript, without using these acronyms, and reserve them for the equations and figure legends. While this would increase manuscript length to some degree, the improved readability would be worth the space.

Line 108, they refer to figure 2A as a “dynamic version of the Goldmann equation”. As presented, it assumes that the reader is thoroughly familiar with this equation. Restating the meaning of the Goldmann equation would make it easier to understand the parallels between the different components of their eye model and the components of the equation. In their description of the apparatus, they refer multiple times to the “pump duty cycle”. This should be defined to improve clarity.

The description of the tubing implantation technique is very cryptic, referring to previously published reports. It is not entirely clear just how the silicone tubing is held in place in order to prevent movement and extrusion from the eye. The statement that they use “half-thickness scleral sutures” is difficult to believe, given that the rat sclera is exceedingly thin and the size suture and needle are not stated. Important features of the outcome of this technique that should be presented include numbers of animals implanted and success rate. Additional factors that are not presented include complications from the implantation itself, including incidence of bleeding and inflammation or infection, and how long these tubes might remain functional. The authors present results from a few animals, but the future utility of this potentially promising method will depend heavily on the reliability of these implantations.

Line 225 refers to a method they have developed to reduce noise in IOP and flow measurements in awake animals “by recursively regressing raw data”. A short explanation of what this does, rather than simply presenting the mathematical equations, would improve understanding of what they are doing here. Why they performed this in every 2 hour or every 6 hour intervals is not clear. Additionally, there is no indication of numbers of animals used for each approach. In line 233, they state that “resting IOP was checked by tonometry”. They do not state if these are done with the animals awake or not. If done awake, how do they account for the possibility that restraining animals for pressure measurement (particularly with tether attached) might result in artificially high pressure readings? They do not mention what the results of these tonometer readings were.

The legend for Figure 4 appears to have an error, in that it lists 4 panels: A,B,C,D, but the text and figure itself have only 3 panels. It appears that what they call panel B (in the legend) should be the right hand portion of panel A. This needs to be corrected. For this figure, and for figure 5 the black and gray traces need to contrast more. They’re very hard to tell apart.

Lines 313, 332 and 544, refer to reported values for facility and various factors that may lead to variability in IOP, respectively. In all cases, the authors just reference their own work. They should include the work of others, such as that of Crawford Downs, who has published telemetry IOP data in monkeys. References to work independent of that of the authors would strengthen the validity of the relevance of values obtained by their current system.

Figure 7 shows interesting data that suggests there may be a diurnal rhythm to conventional outflow facility, with flow and facility being lower at night (dark phase) and higher during the light phase. If true, this would be a new and important finding. As the authors mention, aqueous humor production is known to have a circadian rhythm, being higher during the dark. While the ability to determine flow and facility in awake animals would be unique, the variable rate of aqueous production is a factor that standard aqueous humor dynamics studies do not have to contend with. That said, it is not clear how their algorithm accounts for this factor and that this might not be influencing this result. The presentation of the dark and light phase is not consistent both within figure 7 and between Figure 7A and Figure 8. In the former the dark phase is presented in the middle; elsewhere it is the light phase that is in the middle. The need for this switch is not clear and it makes it more difficult to grasp these results, particularly when looking at these in terms of IOP, flow and facility, some of which are inverse of each other.

Regarding Figure 8, the quality of anterior segment photos should be better. Even with this, the top photo (Y45) shows definite blood subconjunctivally, as well as a clot and likely fibrin on the tube tip, leading to adhesions and pupil distortion. This raises concerns that tube insertion was not without complications and that this and accompanying inflammation could possibly be affecting their readings. The middle photo (Y72), in which the tube is stated to be under the iris, one can see the tube before it goes into the pupil, but there is also several areas of what look like fibrin as well as possible inflammatory debris and some adhesion of the pupil to either the lens surface or the tube. In the bottom image (Y77) it is difficult to actually see the tube but it appears to be visible in the pupil. In fact it may actually be behind the iris here too, and there appears to be evidence of pupil distortion and likely adhesions. Again, one wonders what the effects of these changes might be on their system readings and the longevity of the system as a whole. Line 366 states that “it can be seen that prolonged perfusion had minimal impact on implanted eyes”. Given that in these animals the longest duration of observation was 7 days, this is not particularly prolonged and this statement should be modified. Finally, the statistics applied to these 3 eyes’ data to show a significant difference between dark and light phase is not clear, including just how the n’s equaling the number of days were used and what they consist of.

The Discussion very reasonably points out some of the unique aspects of this system and its potential advantages. While the observation of a diurnal change in facility is potentially interesting, the number of animals it’s demonstrated in here is small, and there are no data presented to suggest that this going to be a consistent finding. Line 433 states that this report shows that “ocular hypertension can be induced in animal models without experimentally damaging outflow pathways.” While this is a potential, the concerns raised above regarding images of Figure 8 would suggest that outflow damage from the surgical implantation of the tube or its presence over a long period of time is a very real possibility. More animals, with documentation of tube failures (success rate) and complications will still be needed in order for this observation to be more convincing. The authors also mention the potential of using this implanted system as a way of delivering therapeutics or performing experimental pharmacologic manipulations. While this is true, it is not clear just how dosage would be standardized or monitored, as the drug would have to first go through the tubing before it gets to the eye, and the rate at which this would occur is unknown. This could certainly be considered as an important area for future development.

6. PLOS authors have the option to publish the peer review history of their article (what does this mean?). If published, this will include your full peer review and any attached files.

Reviewer #1: No

Reviewer #2: No

---

## [Author Response · Author response to Decision Letter 0]

26 Oct 2022

We thank the reviewers for their supportive comments and constructive feedback. Below is our response to points raised.

Reviewer #1 

1. My main concern with this paper is that the authors have not put their data in context. Others have measured outflow facility in the rat eye, including the high-fidelity measurements of Feola et al. and previous measurements by the current authors, plus possibly others that I am not aware of. These should be compared with the data gathered here. 

Response: Feola et al measured facility in Brown Norway rats to be 21.2 [11.2, 40.0] and 26.1 [23.8, 28.7] in young and middle-aged Brown Norway rats, respectively, which is similar to the facility results reported previously by our group and in this paper. The manuscript has been updated to highlight these comparisons in lines 335-336 and 460-466. 

2. Similarly, the flow-pressure graphs (Fig 5B) can be used to get an estimate of aqueous inflow rate by extrapolating the lines back to (estimated) EVP, at which point the (negative) pump flow rate should equal inflow rate. Obviously, there are assumptions underlying such a calculation, but it would be good to see what values the authors get. Measuring inflow rate in small eyes like the rat is very technically challenging and having more data points, with suitable caveats, is useful. 

Response: EVP and uveoscleral outflow are unknown so such a calculation would indeed require assumptions, which we have discussed previously (Ficarotta et al 201X). Y-intercepts were already provided for the regression lines in legends so that the reader can extrapolate flow for any pressure desired. Summary results for y-intercepts of Fig 5D are now provided in the legend as well in line 350.

3. Finally, what is the success rate? The authors show 3 animals that seem to have worked, but how many were tried? I know there was some information on this in previous papers, but now that the authors have more experience, a quantitative update should be given. 

Response: A total of 7 awake free-moving rats were surgically implanted with a cannula and tethered to the device. 4 of these animals yielded pressure and flow data for one or more days, resulting in a success rate of ~60%. The text has been modified to reflect this in lines 353-355 and 536-537.

4. The first two figures provide a good overview of the device size, setup, and use. The compact nature and basic components of the device are clear from figure 1A, and it’s great that this is shown at the start of the paper. I may have liked to see the device with the cannula attached to a conscious rat as well, but this is partially addressed in figure 8. Request: please include a photo in Fig 1 of the device in use with a rat. 

Response: Fig 1 and its legend have updated to include a photo of the experimental setup in action.

5. One thing the authors do not comment on is how bubbles are avoided/dealt with. Bubbles are the bane of all who perfuse eyes, and some comments on this important topic are needed. 

Response: Eliminating any bubbles in the fluidic lines was indeed vital to consistent device operation. This was done by carefully and sequentially filling each component while assembling the device and then stepping the pump to maximum rate multiple times to flush out any bubbles, which was confirmed visually before experimental use. In awake animal experiments, the fluidic line was inspected every 2-3 days to verify that no bubbles have been introduced. If one was found the tether was disconnected, and the line was flushed as described above. The text has been revised to reflect these comments in lines 164-167.

6. The model shown in Figure 2A is useful to demonstrate Goldmann’s equation and visualize the device setup in more detail. It also helps with interpretation of the figures throughout the paper, as readers can reference how each variable in the future graphs relates to the rest of the system. However, I believe there is an error in Fig 2A: the R_G element should not be there. I understand this is to represent viscoelastic globe stretching, but that is not a fluid resistance element per se. Globe stretching can be represented by the compliance C_G. In any case, R_G does not seem to be used anywhere in the paper, so deleting it should not change anything so far as I can tell. 

Response: We acknowledge that an R_G element is not generally included in models of ocular fluid dynamics, which may be why the reviewer considers its inclusion an error. However, we do not believe that globe stretching can be represented solely by compliance C_G because the globe does not instantaneously gain or lose volume following step changes in pressure (Schlegel et al., IOVS, 1972; Collins and van der Wereff, Mathematical models of the dynamics of the human eye, 1980). In a forthcoming manuscript, we report results of experiments that measure its value and include it here for model consistency. The value is small so the reviewer’s belief is not entirely unfounded, but it is not zero.

7. While the assessment of the pump/system parameters is important (Figs 2B-D), I have some concerns. It would help if the dashed line were labeled as nominal load in Fig. 2B. More importantly, the slope of the load line in Fig 2B (and the operating range in Figs 2C and D) are not in the right range. I understand that this refers to the situation before the internal flow limiter is used, but Fig 2C somehow implies that the load that the pump sees in operation is 0.1 mmHg min/ul, which is far from the resistance of the eye (by 1-2 orders of magnitude). Fig 2C is similarly misleading. For Fig 2D, we only care about the very bottom part of the graph, right? The other lines are useless in practice, so a zoomed view of the bottom of the graph (with more data) would be useful. This whole section was confusing and should be reworked/better explained/changed. 

Response: The slope of the line in Fig 2B is arbitrary and chosen for illustration purposes. If it were in the resistance range of the flow limiter or the eye in that figure, the line would be nearly horizontal and that figure would not be terribly informative. Fig 2C is similarly illustrative and shows how Pp would vary for a range of slopes, with the point corresponding to the slope in Fig 2B. The point is not the actual flow limiter resistance. Fig 2D provides actual flow restrictor resistance data that shows Pp is constant independent of restrictor load if it is in the flat range of Fig 2C. The slope and point were labeled “arbitrary load” in Fig 2B and 2C, “load independent regime” marked in 2C (which is main reason for figure), text revised in lines 126-144, and legend revised in lines 154-157 to improve understanding of the figures.

8. Overall, the system validation seems sound (Fig 3), showing the repeatability of the device in a benchtop setup before introducing the noise that results from measuring a live, conscious animal. A minor critique is that in Figure 3B, the measurement points for different days are all on top of each other and mostly indistinguishable, I wonder if colors would be better or if there is another way to visualize this data more clearly. 

Response: Color was added and symbols were shrunk in Fig 3B to better differentiate data from different days.

9. Minor comment: when comparing the new system to the independently measured IOP values in anesthetized animals (Fig. 4) and measuring IOP with the new system in conscious animal (Fig. 5), the IOP as measured by the system is referred to as PE (to match the model in Fig. 2, which is helpful), but in later figures looking at diurnal patterns of IOP in rats (Fig. 6-8), it is labeled as IOP. This should be made consistent. 

Response: The different axis labeling is deliberate. Figs 1-5 introduce and test a novel device, so pressure output of the device is referred to as Pe. Once device performance was validated with experiments on anesthetized animals, it could then be referred to as IOP (as done with the output of any other calibrated IOP-sensing device). This is important for Figs 6-8 because they report novel and interesting initial results about the eye (IOP), as reviewers noted, and not the device (Pe). This is now stated in lines 336-337 for clarity.

10. Additionally, it’s not clear what Figure 5C is adding. It provides information for some of the other animals measured, but it is not easy to distinguish individual lines by the different shapes. It is not explicitly stated that the shapes are different animals, and it’s not easy to see how many are shown. This panel should be improved. Further, Fig. 5D would be better as a box and whisker plot rather than a bar plot (although keeping the individual data points and connecting lines). 

Response: Fig 5C aims to show the relatively linear nature of results across animals over the range of pressures tested to support using the slope to estimate facility. This is now stated in line 333. Symbol size and color has been changed for improved visibility, and the legend now indicates that each represents a different animal in lines 347-349. Fig 5D has also been reformatted as a box-and-whisker plot. 

11. Figure 6 demonstrates the capabilities of the device, especially by showing how well the feedback algorithm functions in a moving, awake animal in 6B. The text claims that the amount of variation shown here in outflow facility across a few hours (23, 27, 28 nl/min*mmHg) is “fairly reproducible” (line 340), but does not cite other studies that justify this claim – see also major comment above. 

Response: The text in line 340 was a qualitative statement about the facility results, which were “fairly reproducible despite the heightened IOP noise in conscious animals”. They do fall within the 95% confidence interval [23.8, 28.7] reported by Feola et al. and have almost the same standard deviation [2.6 vs. 2] reported by Ficarrotta et al. for Brown-Norway rats. As subsequent figures show, some of the variability in repeated measurements in Fig 6 (and thus in the cited studies) may be attributed to circadian rhythmicity. Text was added in lines 466-471 to address this comment as well as the comment above. 

12. The data in figure 7 showing the differences in IOP and facility between night and day is really interesting, and I think it’s a highlight of this paper. It shows how necessary this device is because there’s so much more we can learn about diurnal changes in outflow using this device. However, it is confusing that the setup of Fig. 7A is different from B and C (light/dark intervals are inconsistent). Please fix. Same for the diurnal curves in Fig 8. 

Response: Suggested figure changes have been incorporated.

13. While I appreciate that Figure 8 shows the cannula located in different locations on the rat eye, and that the diurnal patterns shown in 7 are still maintained in these other animals with the cannulas located at different points, it is almost impossible to actually see the cannulas in the pictures of rat eyes in figure 8A. In the middle panel it is completely impossible – presumably some of the cannula was in the AC. Zoomed in views/better labeling are needed. 

Response: The cannula is only visible in the first and last panel. It is completely behind the iris in the middle panel and thus cannot be seen. This is now stated explicitly in the legend in lines 410-412. An inset has also been added to the former panels that zooms in and enhances contrast of the cannula tip to improve visibility.

14. The discussion mentions measuring ocular viscoelasticity (line 441), and although this is included in the system model, data isn’t shown validating this measurement. Suggest just deleting this part.

Response: Suggested text has been deleted.

Reviewer #2: 

15. Generally, this is a difficult manuscript to read. Much of this is due the use of a large number of acronyms, many of which do not appear intuitive, or are inconsistent. For example, CG stands for “globe wall compliance”, whereas C is generally used to indicate facility in some parts of this manuscript and generally in the aqueous humor dynamics literature. Also, line 97, they refer to “PF” mode. It’s not clear what this is, although it appears to be what they earlier define as “CP” (line 64). They persist in using this “PF” designation throughout the manuscript, without ever applying a definition. The authors need to carefully evaluate their acronyms, make sure they are consistent and use them as little as possible in order to minimize the amount of work they are asking of the reader. While a table providing definitions would be somewhat helpful, it would still be extremely cumbersome. A strong suggestion would be, in addition to a table, to spell out the terms in full in the body of the manuscript, without using these acronyms, and reserve them for the equations and figure legends. While this would increase manuscript length to some degree, the improved readability would be worth the space.

Response: We agree that the use of C for facility in aqueous humor literature makes for confusion with compliance (commonly abbreviated C), but then calling it G (since it is conductance) could also be confusing. Per the reviewer’s suggestions, we eliminated use of C in equations, figures, and text and spell out facility everywhere to avoid confusion. We also eliminated the PF acronym, replacing it with CP everywhere for consistency, and spell out OD, ID, and SD to reduce the number of acronyms.

16. Line 108, they refer to figure 2A as a “dynamic version of the Goldmann equation”. As presented, it assumes that the reader is thoroughly familiar with this equation. Restating the meaning of the Goldmann equation would make it easier to understand the parallels between the different components of their eye model and the components of the equation. 

Response: The Goldmann equation definition was added in line 114.

17. In their description of the apparatus, they refer multiple times to the “pump duty cycle”. This should be defined to improve clarity.

Response: The “pump duty cycle” is the percentage of time that the pump is on during a powering cycle. 100% duty cycle would thereby give the maximum flow rate that the pump can output for a given load. This is now clarified in lines 125-126.

18. The description of the tubing implantation technique is very cryptic, referring to previously published reports. It is not entirely clear just how the silicone tubing is held in place in order to prevent movement and extrusion from the eye. The statement that they use “half-thickness scleral sutures” is difficult to believe, given that the rat sclera is exceedingly thin and the size suture and needle are not stated. Important features of the outcome of this technique that should be presented include numbers of animals implanted and success rate. 

Response: The cannula is held in place with 11-0 sutures that are arranged in a z-patten on the sclera and slack is added before attachment to the head mount so that the eye moves freely and any tugging forces from eye or head movement are minimal and transferred to the sclera and do not pull the tip out of the eye. The head mount then isolates the cannula from any tugging from animal movements. The text was modified to give more information in lines 230-234. Success rate of chronically implanted animals was addressed in #3.

19. Additional factors that are not presented include complications from the implantation itself, including incidence of bleeding and inflammation or infection, and how long these tubes might remain functional. The authors present results from a few animals, but the future utility of this potentially promising method will depend heavily on the reliability of these implantations.

Response: We have reported on the cannula implantation method and discussed bleeding and inflammatory issues in our previous manuscripts (e.g. Bello et al 2017, Nicou et al. 2021). The surgical procedures require some skill to perform in rats owing to their small eyes and anterior chamber. They would likely be easier in rabbits or larger animals and more difficult, if not impossible, in mice. Bleeding is rarely an issue but inflammatory responses can be challenging. We are always working to maximize experiment longevity and have added text on the anti-clogging solution used in this study in lines 227-230 and lines 528-538.

20. Line 225 refers to a method they have developed to reduce noise in IOP and flow measurements in awake animals “by recursively regressing raw data”. A short explanation of what this does, rather than simply presenting the mathematical equations, would improve understanding of what they are doing here. 

Response: The method was broken further down in lines 242-253 to improve understanding of how it works, and some insight has been provided into why it produces smooth IOP and flow traces in spite of heightened IOP noise in conscious animals. A manuscript will soon be submitted that describes this method in detail, along with other methods of outflow facility measurement.

21. Why they performed this in every 2 hour or every 6 hour intervals is not clear. Additionally, there is no indication of numbers of animals used for each approach. 

Response: The text now states 4 animals yielded facility data (see #3), so Fig 7C and 8 show that each interval was performed on 2 animals. The two intervals were a tradeoff between temporal resolution and pressure perturbation frequency, which is now mentioned in lines 254-255.

22. In line 233, they state that “resting IOP was checked by tonometry”. They do not state if these are done with the animals awake or not. If done awake, how do they account for the possibility that restraining animals for pressure measurement (particularly with tether attached) might result in artificially high pressure readings? They do not mention what the results of these tonometer readings were.

Response: The text was revised in lines 255-256 to indicate that tonometry measurements were conducted with animals under brief anesthesia. Results of a tonometry check from an implanted animal are provided below. They are not reported because pressure sensor data from our wireless and tethered systems have been validated many times by tonometry in prior manuscripts (eg. Bello et al. 2017; Ficarrotta et al. 2020; Nicou et al. 2021) and we validated the system here in anesthetized animals with an independent pressure sensor and flowmeter, which is far more rigorous a test than tonometry. 

23. The legend for Figure 4 appears to have an error, in that it lists 4 panels: A,B,C,D, but the text and figure itself have only 3 panels. It appears that what they call panel B (in the legend) should be the right hand portion of panel A. This needs to be corrected. For this figure, and for figure 5 the black and gray traces need to contrast more. They’re very hard to tell apart.

Response: The legend of Fig 4 has been corrected in lines 313-323 to match panel lettering and gray traces are now red in Figs 4 and 5 to improve visibility.

24. Lines 313, 332 and 544, refer to reported values for facility and various factors that may lead to variability in IOP, respectively. In all cases, the authors just reference their own work. They should include the work of others, such as that of Crawford Downs, who has published telemetry IOP data in monkeys. References to work independent of that of the authors would strengthen the validity of the relevance of values obtained by their current system.

Response: Additional relevant literature has been cited in lines 335=336, 357-359, and elsewhere. See also #1.

25. Figure 7 shows interesting data that suggests there may be a diurnal rhythm to conventional outflow facility, with flow and facility being lower at night (dark phase) and higher during the light phase. If true, this would be a new and important finding. As the authors mention, aqueous humor production is known to have a circadian rhythm, being higher during the dark. While the ability to determine flow and facility in awake animals would be unique, the variable rate of aqueous production is a factor that standard aqueous humor dynamics studies do not have to contend with. That said, it is not clear how their algorithm accounts for this factor and that this might not be influencing this result. 

Response: For aqueous humor production to influence our results, it would have to change with pump-induced increases in IOP so as to conflate with a facility change. We are unaware of evidence for this. The Goldmann equation, as customarily written, would also require modification if this were the case. Diurnal variations in aqueous humor production occur on much too slow of a time scale to have altered outflow significantly over the 30-40 min duration of each facility measurement. Other evidence that production is constant during facility measurements is that IOP returns to its resting value afterwards in Figs 6A and 7A. The text has been modified to further clarify these points in lines 445-452.

26. The presentation of the dark and light phase is not consistent both within figure 7 and between Figure 7A and Figure 8. In the former the dark phase is presented in the middle; elsewhere it is the light phase that is in the middle. The need for this switch is not clear and it makes it more difficult to grasp these results, particularly when looking at these in terms of IOP, flow and facility, some of which are inverse of each other.

Response: The suggested figure changes have been incorporated.

27. Regarding Figure 8, the quality of anterior segment photos should be better. Even with this, the top photo (Y45) shows definite blood subconjunctivally, as well as a clot and likely fibrin on the tube tip, leading to adhesions and pupil distortion. This raises concerns that tube insertion was not without complications and that this and accompanying inflammation could possibly be affecting their readings. The middle photo (Y72), in which the tube is stated to be under the iris, one can see the tube before it goes into the pupil, but there is also several areas of what look like fibrin as well as possible inflammatory debris and some adhesion of the pupil to either the lens surface or the tube. In the bottom image (Y77) it is difficult to actually see the tube but it appears to be visible in the pupil. In fact it may actually be behind the iris here too, and there appears to be evidence of pupil distortion and likely adhesions. Again, one wonders what the effects of these changes might be on their system readings and the longevity of the system as a whole. Line 366 states that “it can be seen that prolonged perfusion had minimal impact on implanted eyes”. Given that in these animals the longest duration of observation was 7 days, this is not particularly prolonged and this statement should be modified.

Response: The statement has been deleted as clearly what is prolonged and minimal to us may not be to others.

28. Finally, the statistics applied to these 3 eyes’ data to show a significant difference between dark and light phase is not clear, including just how the n’s equaling the number of days were used and what they consist of.

Response: A t-test was performed on daytime (6A-6P) measurements against nighttime (6P-6A) measurements across all days for each animal. The text was modified to clarify this in lines 396-399.

29. The Discussion very reasonably points out some of the unique aspects of this system and its potential advantages. While the observation of a diurnal change in facility is potentially interesting, the number of animals it’s demonstrated in here is small, and there are no data presented to suggest that this going to be a consistent finding. 

Response: Although the N is indeed small, every animal showed a facility rhythm and such a rhythm has been reported in rabbits and humans with fluorophotometry. The text has been modified to include the fluorophotometry findings and citations in lines 438-440. 

30. Line 433 states that this report shows that “ocular hypertension can be induced in animal models without experimentally damaging outflow pathways.” While this is a potential, the concerns raised above regarding images of Figure 8 would suggest that outflow damage from the surgical implantation of the tube or its presence over a long period of time is a very real possibility. More animals, with documentation of tube failures (success rate) and complications will still be needed in order for this observation to be more convincing. 

Response: This statement was intended to convey that other ocular hypertension models achieve IOP elevation by obstructing aqueous outflow pathways, oftentimes permanently, while the device presented is capable of reversibly elevating IOP without need for explicitly damaging the outflow pathway. The text has been modified to clarify this in line 479-480. 

31. The authors also mention the potential of using this implanted system as a way of delivering therapeutics or performing experimental pharmacologic manipulations. While this is true, it is not clear just how dosage would be standardized or monitored, as the drug would have to first go through the tubing before it gets to the eye, and the rate at which this would occur is unknown. This could certainly be considered as an important area for future development.

Response: Drug delivery is mainly possible because of the removable tether system. With only 3-5 ul of dead volume before the eye, a small aspiration from the pinport to avoid fluid overload and subsequent drug injection would allow for drug delivery by diffusion or pump infusion directly into the anterior chamber. The text has been modified to communicate this in lines 484-485.

---

## [Decision Letter · Decision Letter 1]

12 Dec 2022

PONE-D-22-19808R1A portable feedback-controlled pump for monitoring eye outflow facility in conscious ratsPLOS ONE

Dear Dr. Passaglia,

Thank you for submitting your manuscript to PLOS ONE. After careful consideration, we feel that it has merit but does not fully meet PLOS ONE’s publication criteria as it currently stands. Therefore, we invite you to submit a revised version of the manuscript that addresses the points raised during the review process.

We look forward to receiving your revised manuscript.

Kind regards,

Ted S Acott, PhD

Academic Editor

PLOS ONE

Journal Requirements:

Additional Editor Comments (if provided):

One reviewer had minor suggestions that would greatly strengthen this important paper - please address them

Reviewers' comments:

Reviewer's Responses to Questions

**Comments to the Author**

1. If the authors have adequately addressed your comments raised in a previous round of review and you feel that this manuscript is now acceptable for publication, you may indicate that here to bypass the “Comments to the Author” section, enter your conflict of interest statement in the “Confidential to Editor” section, and submit your "Accept" recommendation.

Reviewer #1: (No Response)

Reviewer #2: All comments have been addressed

2. Is the manuscript technically sound, and do the data support the conclusions?

Reviewer #1: Yes

Reviewer #2: Yes

3. Has the statistical analysis been performed appropriately and rigorously? 

Reviewer #1: Yes

Reviewer #2: Yes

4. Have the authors made all data underlying the findings in their manuscript fully available?

Reviewer #1: Yes

Reviewer #2: Yes

5. Is the manuscript presented in an intelligible fashion and written in standard English?

Reviewer #1: Yes

Reviewer #2: Yes

6. Review Comments to the Author

Reviewer #1: The authors have been generally responsive and the manuscript is improved. However, there are still a few concerns as follows.

1. R_G element in Figure 2A. We asked that this be removed but the authors wanted to retain it, stating that it must be present because "the globe does not instantaneously gain or lose volume following step changes in pressure". While it is true that the globe does not instantaneously change in volume, the inference that R_G is required to explain this behavior is categorically false. More specifically, the globe compliance interacting with the outflow resistance and/or needle resistance creates a RC time constant that can explain globe filling dynamics. The only reason I can imagine to include R_G is to represent the viscoelastic response of the corneoscleral shell, in which case I guess the shell is being represented as a Maxwell body. I doubt the system has the sensitivity to measure this. Rather than having this extremely confusing element in the diagram which is never used, why not just delete it???

2. Suggest moving Fig. 2B-C to supplemental. Characterization of the pump with arbitrary load is not essential for readers to understand the paper and is confusing given that the load shown is not relevant to aqueous humor drainage in rats. It may be useful for a small subset of people interested in the specific pump dynamics, but appearing in the supplemental material is fine for that purpose.

Reviewer #2: the authors have adequately addressed my initial comments

7. PLOS authors have the option to publish the peer review history of their article (what does this mean?). If published, this will include your full peer review and any attached files.

Reviewer #1: No

Reviewer #2: No

---

## [Author Response · Author response to Decision Letter 1]

21 Dec 2022

We thank the reviewers for examining the manuscript a second time. Reviewer #2 has no further comments. Below is our response to the few remaining concerns raised by Reviewer #1.

Reviewer #1 

1. R_G element in Figure 2A. We asked that this be removed but the authors wanted to retain it, stating that it must be present because "the globe does not instantaneously gain or lose volume following step changes in pressure". While it is true that the globe does not instantaneously change in volume, the inference that R_G is required to explain this behavior is categorically false. More specifically, the globe compliance interacting with the outflow resistance and/or needle resistance creates a RC time constant that can explain globe filling dynamics. The only reason I can imagine to include R_G is to represent the viscoelastic response of the corneoscleral shell, in which case I guess the shell is being represented as a Maxwell body. I doubt the system has the sensitivity to measure this. Rather than having this extremely confusing element in the diagram which is never used, why not just delete it???

Response: We have removed the R_G element from the model in Figure 2A. 

2. Suggest moving Fig. 2B-C to supplemental. Characterization of the pump with arbitrary load is not essential for readers to understand the paper and is confusing given that the load shown is not relevant to aqueous humor drainage in rats. It may be useful for a small subset of people interested in the specific pump dynamics, but appearing in the supplemental material is fine for that purpose.

Response: We have moved Fig 2B-2D and the accompanying text on pump operation and how it was made into a constant pressure source to a new Supplemental Information section (S1 Appendix).

---

## [Editor Report · Decision Letter 2]

27 Dec 2022

A portable feedback-controlled pump for monitoring eye outflow facility in conscious rats

PONE-D-22-19808R2

Dear Dr. Passaglia,

We’re pleased to inform you that your manuscript has been judged scientifically suitable for publication and will be formally accepted for publication once it meets all outstanding technical requirements.

Kind regards,

Ted S Acott, PhD

Academic Editor

PLOS ONE
---

## [Editor Report · Acceptance letter]

3 Jan 2023

PONE-D-22-19808R2 

A portable feedback-controlled pump for monitoring eye outflow facility in conscious rats 

Dear Dr. Passaglia:

I'm pleased to inform you that your manuscript has been deemed suitable for publication in PLOS ONE. Congratulations! Your manuscript is now with our production department. 

Kind regards, 

on behalf of

Dr. Ted S Acott 

Academic Editor

PLOS ONE